# D-Pinitol Improved Glucose Metabolism and Inhibited Bone Loss in Mice with Diabetic Osteoporosis

**DOI:** 10.3390/molecules28093870

**Published:** 2023-05-04

**Authors:** Xinxin Liu, Tomoyuki Koyama

**Affiliations:** Department of Marine Bioscience, Tokyo University of Marine Science and Technology, 4-5-7 Konan, Minato City, Tokyo 1080075, Japan; d221020@edu.kaiyodai.ac.jp

**Keywords:** D-pinitol, diabetic osteoporosis, D-chiro-inositol, GC-MS, mice

## Abstract

Diabetic osteoporosis (DO) has been increasingly recognized as an important complication of diabetes. D-pinitol, a natural compound found in various legumes, is known for its anti-diabetic function, but its effect on DO has not been investigated. Two doses of pinitol (50 and 100 mg/kg Bw/d) were administered orally to experimentally induce the DO mouse model for 5 weeks. The results indicated that pinitol suppressed fasting blood glucose levels and tended to enhance impaired pancreatic function. Pinitol also suppressed serum bone turnover biomarkers, and improved dry femur weight, cancellous bone rate, and bone mineral content in the DO mice. Based on the inositol quantification using GC-MS in serum, liver, kidney, and bone marrow, the pinitol treatment significantly recovered the depleted D-chiro-inositol (DCI) content or the decreased the ratio of DCI to myo-inositol caused by DO. In short, our results suggested that pinitol improved glucose metabolism and inhibited bone loss in DO mice via elevating the DCI levels in tissues.

## 1. Introduction

According to the International Diabetes Federation, about 451 million individuals are affected by diabetes mellitus (DM) globally. It is predicted that the number of diabetic people will increase to 693 million by 2045 [1]. Meanwhile, more than 200 million people are suffering from osteoporosis. One in three women over the age of 50 years will experience osteoporotic fractures in their lifetime [2]. The prevalence of both diabetes and osteoporosis rises with age and they are quite often coexisting in the elderly. There is ample evidence that osteopenia appears in both type 1 and type 2 diabetic patients [3,4]. Continuous osteopenia eventually results in osteoporosis, which is referred to as diabetic osteoporosis (DO). With the increasing morbidity of DM in the world, the population of patients with DO has increased accordingly [5]. DO is one of the severe complications of diabetes, featuring the decline in bone mineral density due to the long-term abnormal metabolism of nutrients and enhanced inflammatory cytokines [6,7]. Osteoblasts and osteoclasts are responsible for bone formation and resorption, respectively [8]. This dynamic equilibrium keeps the bone renewing all the time. Diabetes affects bone metabolism and strength by influencing osteoblasts and osteoclasts, ultimately causing bone loss [9]. Notably, there are significantly lower estrogen levels in patients with DM [10]. Estrogen deficiency naturally stimulates osteoclast formation, inducing osteopenia to develop to a higher degree. The optimal therapies for DO include hormone replacement therapy (HRT), bisphosphonates, antiresorptive drugs, osteoanabolic drugs, and calcium and vitamin D (VD) supplements [11,12]. However, given that none of these therapies are effective in controlling high blood glucose, it is imperative to find a harmless and effective substance with a hypoglycemic effect for DO treatment.

D-pinitol, a 3-methoxy analogue of D-chiro-inositol (DCI), was first isolated and structurally characterized from the pine tree [13]. It is a prominent component in a variety of legumes, especially in *Ceratonia siliqua* L. (English name: carob) [14]. It has been widely assumed that D-pinitol possesses multiple bioactivities, including anti-cancer [15], anti-diabetes [16,17], antioxidant [18], hepatoprotective [19], and anti-osteoporosis [20] bioactivities. The most notable bioactivity of pinitol is its insulin-like effect. Pinitol exerts its anti-hyperglycemic activity via promoting the expression of PI3K as well as the downstream target Akt to accelerate glycogen synthesis in rats with type 2 diabetes [16]. Furthermore, there is a lot of research suggesting that pinitol has a positive effect on various diabetic complications, such as anti-hyperlipidemia [17], reducing cardiovascular risk [21], and restoring the impaired activities of hepatic key enzymes [22].

Compared to the massive evidence for the anti-diabetes properties of pinitol, only a few researchers have reported the effect of pinitol on bone metabolism. It has been reported that DCI negatively regulates the formation of osteoclasts by down-regulating the nuclear factor of activated T cells c1 (NFATc1) [23]. This implies the potential for pinitol in osteoporosis treatment, since convincing evidence suggests that D-pinitol can be actively converted to DCI in cells in the body [23,24]. D-pinitol has been demonstrated to be capable of inhibiting RANKL-induced osteoclastogenesis [20]. However, pinitol has not been shown to have any significant effect on osteoblasts. Furthermore, the anti-osteoporotic effect of pinitol has been proved in ovariectomized mice, with increases in bone mineral density and bone mineral content, while C-terminal telopeptides decrease significantly after 5 weeks of pinitol treatment. However, the effects of pinitol treatment on the DCI content in tissues have not yet been reported mainly due to its extremely low content in tissues. Compared to expensive DCI, the wide distribution of pinitol in plants probably makes it affordable for people for osteoporosis prevention and treatment.

To evaluate the protective effects of pinitol in DO and clarify its possible mechanism, a DO mouse model was established using a streptozotocin (STZ) injection coupled with an ovariectomy. The metabolic biomarkers for glucose and bone, the physical characteristics of bone, and the distribution of pinitol and its derivatives were investigated in DO mice after 5 weeks of pinitol treatment.

## 2. Results

### 2.1. Uterus Weight

As shown in Figure 1, the uterine weight of the Sham and DO mice differed significantly (*p* < 0.001), while the pinitol treatment at each dose had no significant effect on uterine weight.

### 2.2. Body Weight and Food Intake

The body weight of the DO group decreased significantly when compared with the Sham group, whereas pinitol at both the low dose and high dose had no significant effect on mice body weight (Figure 2A). As shown in Figure 2B, the food intake in the DO mice was significantly higher than that in the Sham group, while pinitol had no significant effect.

### 2.3. Fasting Blood Glucose Levels and Insulin Metabolism

As shown in Figure 3A, the FBG in the DO group increased dramatically within the experiment when compared with the Sham group. However, pinitol suppressed the increase in the FBG from the third week in either dose. At the end of the experiment, the FBG in both the low dose and high dose pinitol-treated mice was significantly lower than in the untreated mice (*p* < 0.05). Furthermore, the Fins in the DO group were significantly lower than in the Sham group (*p* < 0.05) due to the impaired pancreatic function induced by STZ. However, the Fins increased dose-dependently after the pinitol treatment (Figure 3B). A similar trend was observed in the results of the HOMA-β (Figure 3C), implying the recovery of pancreatic function after pinitol treatment. However, the HOMA-IR of each group had no significant difference (Figure 3D).

### 2.4. Serum Bone Biomarkers

As shown in Table 1, there was no significant difference in serum calcium among the groups; however, a high dose of pinitol treatment significantly increased the serum phosphorus levels when compared with the DO group (*p* < 0.05). To evaluate the effects of pinitol on the bone metabolic state in the mice, the serum BALP activity and TRAcP activity was measured. The serum BALP activity was used as a marker of bone formation, and TRAcP activity was used as a marker for bone resorption. The serum BALP and TRAcP activity increased to some extent, implying an increase in bone turnover, whereas both tended to decrease after the pinitol treatments.

### 2.5. Bone Parameters

As shown in Figure 4B,C, ovariectomy significantly decreased the dry femur weight (*p* < 0.01) and cancellous bone rate (*p* < 0.001), and the decline in the cancellous bone microstructure can also be observed in Figure 4A. The pinitol treatment significantly inhibited the loss of cancellous bone dose-dependently, which was in agreement with the presented cross-sectional view in Figure 4A. In addition, a high dose of pinitol treatment significantly increased the Ca and P content in femur ash (Figure 4D) (*p* < 0.05). Moreover, the ratio of Ca/P in the femur ash of each group did not show a significant difference (Figure 4E).

### 2.6. Inositol Content

The MI and DCI contents were measured using GC-MS as shown in Table 2. We could not detect pinitol in any tissue due to the detection limit. The DCI contents or DCI to MI ratio in the DO group decreased significantly in all tissues (*p* < 0.05), except in the bone marrow when compared with the Sham group, while there was no significant difference in the MI content in all tissues among the groups. A high dose of pinitol treatment remarkably elevated the ratio of DCI to MI in serum, kidney, and bone marrow protein in the DO mice when compared with the untreated group (*p* < 0.05) in a dose-dependent manner. Even if there was no statistical difference, the DCI contents and ratio of DCI to MI in the liver also tended to increase after being treated with either dose of pinitol for 5 weeks.

## 3. Discussion

Pinitol is widely accepted as an anti-diabetic ingredient in food. In our study, the effects of pinitol on DO mice were investigated after 5 weeks of treatment. The DO mice showed significantly lower uterine weight and significantly higher FBG at the end of the experiment, suggesting that our construction of the animal models was successful. After 5 weeks of pinitol treatment, the FBG in the DO mice treated with a low dose and high dose of pinitol decreased 44.34% and 38.99%, respectively, and the impaired pancreatic function also tended to recover, which suggested the improvement of glucose metabolism in the DO mice (Figure 3A). Pinitol suppressed FBG in the DO mice probably via two ways: (1) by activating the phosphatidylinositol-3-kinase/protein kinase B (PI3K/Akt) signaling pathway to promote insulin-mediated glucose uptake [16], (2) by enhancing glucose-mediated insulin secretion [25]. Given that the DO mice had impaired pancreatic function but no insulin resistance, the suppression of FBG in the DO mice was considered to be related to an improvement in pancreatic function (Figure 3C).

In this study, elevated activity of serum TRAcP and BALP, which are biomarkers of bone resorption and formation, respectively, were observed in the DO mice (Table 1). The increased bone formation marker was supposed to be the compensation of the osteoblasts for bone loss in the DO mice [26]. A weak decreasing tendency of serum TRAcP and BALP activity (Table 1), a significant increase in dry femur weight, an increase in cancellous bone weight, and an increase in bone mineral content (Figure 3) indicated the protective effect of pinitol on bone loss in the DO mice. The improvement in glucose and bone metabolism was considered to be highly related to inositol homeostasis.

The DCI contents and the ratio of DCI to MI decreased significantly in the serum, liver, and kidney, while they increased significantly after being treated with a high dose of pinitol for 5 weeks (Table 2). Even if the ratio of DCI to MI in the bone marrow did not decrease significantly in the DO group when compared with the Sham group, they increased dose-dependently after the pinitol treatment. A decline in the DCI levels in muscle, hemodialysate, urine, kidney, and liver from diabetes has been observed by many researchers [27,28,29]. Until the present time, it has been widely acknowledged that the deficit in DCI correlates with insulin resistance, which was explained by an impaired conversion of MI to DCI in GK rats [29]. Yu Jungeun et al. concluded that DCI acted as an inhibitor of RANKL-induced osteoclast differentiation by down-regulating the nuclear factor of activated T cells c1 (NFATc1) through the inhibition of NF-κB in vitro [23]. Furthermore, it has been reported that pinitol inhibited osteoclastogenesis in a similar manner as pinitol-attenuated NF-κB activation [20]. Moreover, both pinitol and DCI manifested an anti-diabetic action via similar mechanisms. On the one hand, pinitol alleviated insulin resistance through the PI3K/Akt signaling pathway in type 2 diabetic rats [16,17], and it also activated the PI3K/Akt signaling pathway in the rat hypothalamus after acute oral administration [30]. On the other hand, DCI also showed a hypoglycemic effect in type 2 diabetic rats through the PI3K/Akt signaling pathway [16,17]. This substantial consistency between DCI and pinitol led us to believe that the conversion to DCI was necessary for pinitol to exhibit the anti-diabetes and anti-osteoporosis activity. Thus, administering DCI is probably an ideal way to bypass the defective epimerization of MI to DCI to at least partially restore inositol homeostasis and improve diabetic osteoporosis. However, the narrow distribution of DCI in nature suggests that it would be costly to use in therapy. In contrast, D-pinitol, the 3-*O*-methyl form of DCI, is much more abundant in Leguminosae plants [31]. Here, we demonstrated that pinitol treatment could effectively increase DCI levels in specific tissues (liver, kidney, and bone), bringing with it the improvement in glucose and bone metabolism.

As one of the complications of diabetes, the incidence of osteoporosis in diabetic patients is over 60% [32]. Previous research suggested that pinitol could effectively improve diabetes, and other researchers also reported the positive effects of pinitol on alleviating osteoporosis caused by estrogen deficiency [16,17,20]. However, there was no evidence that pinitol supplementation was beneficial for DO mice. Moreover, the feasibility of replacing DCI with more accessible pinitol has been rarely discussed. Our results indicated that the elevated DCI levels in serum, liver, kidney, and bone marrow contributed significantly to suppressing FBG and bone loss in DO mice after 5 weeks of pinitol treatment.

## 4. Material and Methods

### 4.1. Material

D-pinitol (>95% purity) was provided by Nihon Advanced Agri Co., Ltd. (Nagahama, Shiga, Japan).

### 4.2. Animals

Female ICR mice (3 weeks old, 18 ± 2 g) were purchased from SLC Inc. (Hamamatsu, Shizuoka, Japan) and were acclimated to laboratory conditions for 1 week before the experiment. The experimental mice were housed in an air-conditioned room at 23 ± 2 °C with 12 h/12 h light–dark illumination cycles at constant humidity (55 ± 10%) and were free to consume distilled water and standard ingredient chow without restriction. All procedures were reviewed and approved in consideration of animal welfare by the Animal Ethics Committee at Tokyo University of Marine Science and Technology (R4-2).

### 4.3. Animal Models and Treatment

Diabetic osteoporosis mouse models were established using intraperitoneal STZ (Fujifilm Wako Pure Chemical Corporation, Osaka, Japan) injection and bilateral ovariectomy. In short, the mice underwent either bilateral laparotomy (Sham, *n* = 7) or bilateral ovariectomy, which were performed under combination anesthesia [33]. After 2 days’ recovery, the mice with ovariectomy were intraperitoneally injected with STZ (110 mg/kg Bw, dissolving in 0.1 M citric acid buffer, pH = 4.5) after 16 h of fasting, while the Sham group mice were injected with only citrate vehicle. One week after STZ injection, venous blood was collected from tails to examine the blood glucose levels with a Glucose C II Test kit (Fujifilm Wako Pure Chemical Corporation, Osaka, Japan). The mice with glucose levels over 250 mg/dL were considered to have diabetic osteoporosis and were selected for further studies. Two days after blood glucose level determination, the mice with diabetic osteoporosis were divided into three groups: diabetic osteoporosis mice orally administered by intragastric gavage with vehicle (DO, *n* = 7), low dose of pinitol (DO + LP, 50 mg/kg Bw, *n* = 7) once per day, high dose of pinitol (DO + HP, 100 mg/kg Bw, *n* = 7) once per day. The Sham group was orally administrated with the vehicle. Pinitol was dissolved in distilled water. The treatments lasted for 5 weeks. The fasting blood glucose (FBG) levels were determined at 0, 2, 4, and 5 weeks from pinitol treatments. Food intake was recorded daily, and the body weights (Bw) were measured once per week.

At the end of the fifth week, all mice were sacrificed under isoflurane anesthesia after 16 h of fasting. Blood samples were taken using cardiac puncture, then centrifuged for 5 min (8000 rpm, 4 °C) to collect serum. Thereafter, uteruses, liver, kidneys, bilateral tibias, and femurs were dissected. Both ends of the tibias and femurs were removed, and the bone marrow cells were collected using centrifugation (4 °C, 8000 rpm, 5 min). All collected samples were stored at −80 °C immediately before analysis.

### 4.4. Assay for Serum Biochemistry

FBG levels were measured using a Glucose C II Test kit (Fujifilm Wako Pure Chemical Corporation, Osaka, Japan). Fasting insulin levels (Fins) were measured using an Ultra-Sensitive Mouse Insulin ELISA kit (Morinaga Institute of Biological Science, Inc., Yokohama, Kanagawa, Japan). Serum phosphorus (P) levels were detected as reported in Taussky et al. [34]. Serum calcium (Ca) levels were detected using *O*-cresolphthalein complexone (OCPC) methods and measured the absorbance at 590 nm using a microplate reader (Thermo Fisher Scientific, Waltham, MA, USA). Serum tartrate-resistant acid phosphatase (TRAcP) activities were measured as reported in Lau et al. [35]. Serum bone-specific alkaline phosphatase (BALP) activities were measured using a ready-to-use 4-nitrophenyl phosphate solution (Tokyo Chemical Industry, Tokyo, Japan) mixed with 10 mM L-phenylalanine as reported in Dimai et al. [36].

### 4.5. Bone Physical Parameters and Bone Mineral Content

Freshly isolated femurs were weighed using an electronic scale after drying at 65 °C for 48 h. Femur cross-sectional views were taken using the macro mode of a Tough TG-1-megapixel camera (Olympus, Tokyo, Japan) after abrasion with a whetstone in a fixed direction and degree. The cancellous bone rate was defined as the rate of the cancellous bone length to the longitudinal length of the entire femur measured using an electronic digital caliper. Bone mineral content (calcium and phosphorous) in ash was determined using the same method as used for serum Ca and P measurement after the femur was ashed at 680 °C for 18 h and subsequently dissolved in 6 N HCl.

### 4.6. Inositol Content in Tissues

Liver, kidney, and bone marrow proteins added with distilled water (m:v = 1:4) were homogenized. The protein contents were measured using a kit (Fujifilm Wako Pure Chemical Corporation, Osaka, Japan). Tissue homogenate or serum (200 μL) was used for determining myo-inositol (MI), DCI, and pinitol contents according to the methods described in Guo J. et al., with slight modifications [37]. A total of 200 μL serum or tissue homogenate were hydrolyzed in 400 μL HCl (6 N) in sealed glass tubes at 100 °C for 24 h. After hydrolysis, all aliquots were transferred to 20 mL vials. The original tubes were washed with ethanol and combined with aliquots. Then, 5 mL ethanol was added to each vial to remove water through evaporation. The residue was mixed with 1.5 mL derivatization reagent (trimethylsilylimidazole: pyridine = 1:4) and dried at 100 °C for 15 min. After cooling to room temperature, the mixtures were transferred to 5 mL of saturated sodium chloride solution. The resulting vial was washed twice with 1 mL hexane and the combined hexane was also transferred to the saturated sodium chloride solution. After vortex and centrifuge (5000 rpm, 5 min), 1 mL supernatant was evaporated until it was dry and reconstituted with 200 μL anhydrous pyridine for GC-MS analysis.

Five microliters of aliquots were injected to a GCMS-QP2010Plus (Shimadzu, Kyoto, Japan) in splitless mode, coupled with an Rtx-5MS capillary column (30 m × 0.25 mm, 0.25 μm, Restek, Bellefonte, PA, USA). Helium was used as the carrier gas at a constant flow rate of 1.5 mL/min. The injector and source temperatures were maintained at 240 and 200 °C, respectively. The column temperature program consisted of injection at 150 °C, which was first increased to 190 °C at 5 °C/min, and then increased to 240 °C at 25 °C/min after holding at 190 °C for 5 min. This was followed by an isothermal hold at 240 °C for 3 min. The mass spectrometer was operated in the electron impact mode with an ionization energy of 70 eV. Pinitol, MI, and DCI were identified by comparing their mass spectrums with their commercial standards [38]. The fragment ion (*m*/*z* 305) was used for quantification in the selected ion monitor (SIM) mode. Serum inositol content was expressed as μg/mL serum and inositol contents in liver, kidney, and bone marrow were expressed as μg/mg protein.

### 4.7. Statistical Analysis

All data were presented as mean ± S.E.M. The significance of differences between experimental groups was evaluated using Student’s *t* test and considered significant at *p* < 0.05. For the statistical analysis of inositol content, the significance of differences between groups was determined using a one-way analysis of variance. The differences were further evaluated using Tukey’s multiple comparison tests and considered significant at *p* < 0.05. All statistical analysis was performed using SPSS 22.0 (SPSS Inc., Chicago, IL, USA).

## 5. Conclusions

This research was conducted to investigate the effects of pinitol on mice with diabetic osteoporosis. In conclusion, pinitol exerted a potent protective effect on mice with diabetic osteoporosis, as 5 weeks of pinitol treatment suppressed FBG, increased dry femur weight, cancellous bone rate, and bone mineral content significantly, which supposedly resulted from the elevated level of DCI content in serum, liver, kidney, and bone marrow. Therefore, our results suggested that the daily intake of pinitol could be an effective strategy for diabetic osteoporosis treatment by refilling the depleted DCI in tissues.

## Figures and Tables

**Figure 1 molecules-28-03870-f001:**
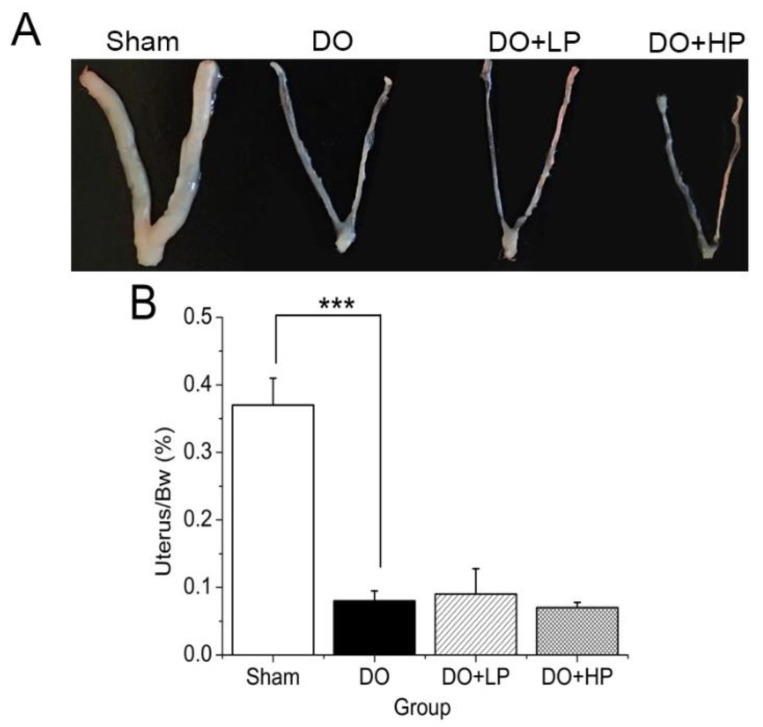
Effect of pinitol on the changes in uterus weight. The Sham group mice were treated with the vehicle (Sham). The mice with diabetic osteoporosis were treated with vehicle (DO), 50 mg pinitol/ kg Bw/day (DO + LP), and 100 mg pinitol/kg Bw/day (DO + HP) for 5 weeks, respectively. (**A**) Photo of the uterus at the end of the experiment. (**B**) Uterus weight over body weight. Values were expressed as mean ± SEM, *n* = 7. *** *p* < 0.001.

**Figure 2 molecules-28-03870-f002:**
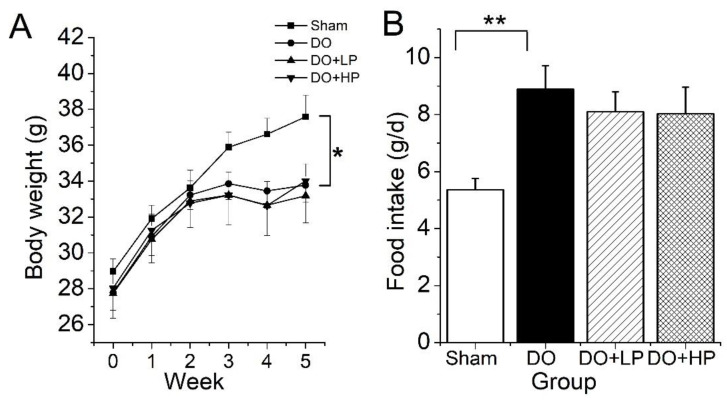
Effect of pinitol on the changes in body weight and food intake during experimental period. The Sham group mice were treated with the vehicle (Sham). The mice with diabetic osteoporosis were treated with the vehicle (DO), 50 mg pinitol/kg Bw/day (DO + LP), and 100 mg pinitol/kg Bw/day (DO + HP) for 5 weeks, respectively. (**A**) Body weight. (**B**) Food intake. Values were expressed as mean ± SEM, *n* = 7. * *p* < 0.05, ** *p* < 0.01.

**Figure 3 molecules-28-03870-f003:**
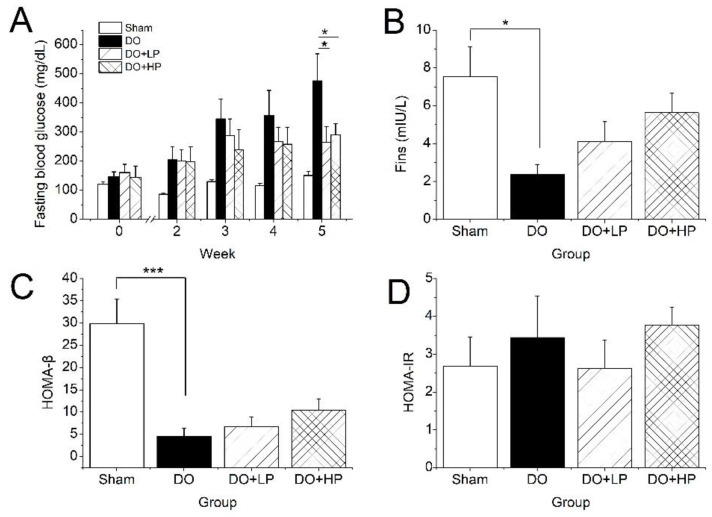
Effect of pinitol on the changes in fasting blood glucose levels during the experimental period and insulin metabolism at the end of the experiment. The Sham group mice were treated with the vehicle (Sham). The mice with diabetic osteoporosis were treated with the vehicle (DO), 50 mg pinitol/kg Bw/day (DO + LP), and 100 mg pinitol/kg Bw/day (DO + HP) for 5 weeks, respectively. (**A**) Fasting blood glucose levels at 0, 2, 3, 4, 5 weeks of pinitol treatment. (**B**) Fasting insulin levels at the end of this experiment. (**C**) HOMA-β. (**D**) HOMA-IR. Values were expressed as mean ± SEM, *n* = 7. * *p* < 0.05, *** *p* < 0.001.

**Figure 4 molecules-28-03870-f004:**
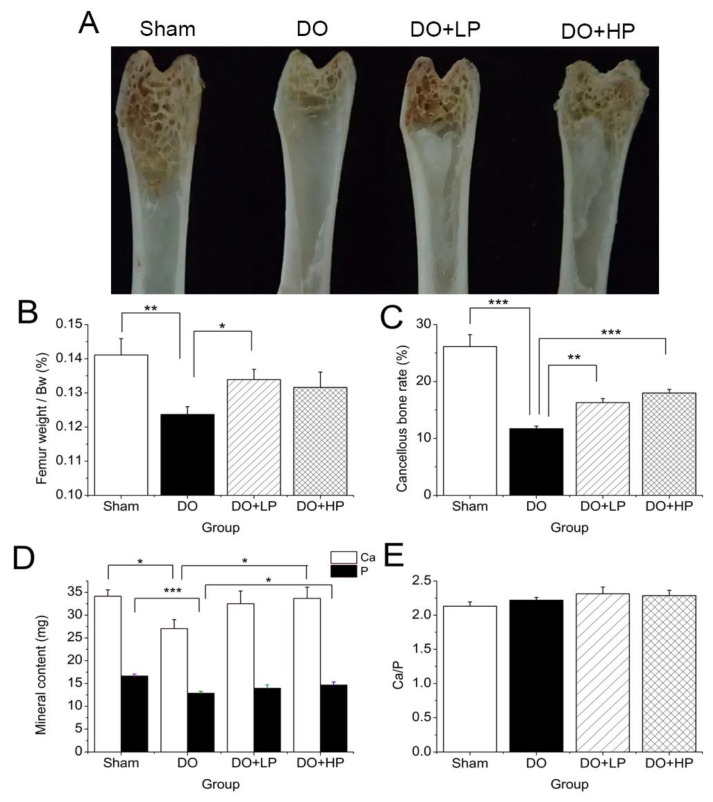
Effect of pinitol on the physical changes in bone and bone mineral content in mice with diabetic osteoporosis. (**A**) Cross-sectional view of the femur at killing. (**B**) Dry femur weight compared to body weight at killing. (**C**) Cancellous bone rate (the longitudinal length of internal cancellous bone compared to total length). (**D**) Calcium content fixed to the left axis and phosphorus content fixed to the right axis compared to ash. (**E**) Ratio of calcium to phosphorus. The Sham group mice were treated with vehicle (Sham). The mice with diabetic osteoporosis were treated with vehicle (DO), 50 mg pinitol/kg Bw/day (DO + LP), and 100 mg pinitol/kg Bw/day (DO + HP) for 5 weeks, respectively. Values were expressed as mean ± SEM, *n* = 7. * *p* < 0.05, ** *p* < 0.01, *** *p* < 0.001.

**Table 1 molecules-28-03870-t001:** Serum parameters.

	Sham	DO	DO + LP	DO + HP
Ca (mM)	2.82 ± 0.03	2.92 ± 0.07	2.91 ± 0.09	2.88 ± 0.04
P (mM)	2.80 ± 0.11	2.99 ± 0.12	3.52 ± 0.18	3.62 ± 0.29 ^#^
TRAcP (U/L)	9.32 ± 0.58	10.02 ± 0.89	9.40 ± 1.05	7.88 ± 0.84
BALP (U/L)	51.10 ± 4.92	81.04 ± 11.34 *	88.66 ± 8.92	80.72 ± 12.15

The Sham group mice were treated with the vehicle (Sham). The mice with diabetic osteoporosis were treated with the vehicle (DO), 50 mg pinitol/kg Bw/day (DO + LP), and 100 mg pinitol/kg Bw/day (DO + HP) for 5 weeks, respectively. Values were expressed as mean ± SEM, *n* = 7. * *p* < 0.05, vs. Sham; ^#^
*p* < 0.05 vs. DO.

**Table 2 molecules-28-03870-t002:** Effect of pinitol on inositol content in diabetic osteoporosis mice.

		Sham	DO	DO + LP	DO + HP
Serum ^$^	DCI	0.215 ± 0.020	0.129 ± 0.017 **	0.156 ± 0.020	0.156 ± 0.021
	MI	76.889 ± 3.971	88.320 ± 8.445	77.517 ± 8.659	73.217 ± 3.321
	DCI/MI (%)	0.316 ± 0.014	0.181 ± 0.014 **	0.238 ± 0.026	0.239 ± 0.041 ^#^
Liver	DCI	0.064 ± 0.015	0.030 ± 0.002 *	0.033 ± 0.008	0.041 ± 0.009
	MI	3.785 ± 0.464	3.314 ± 0.207	3.310 ± 0.178	3.763 ± 0.253
	DCI/MI (%)	1.920 ± 0.569	0.879 ± 0.101 *	1.050 ± 0.296	1.104 ± 0.227
Kidney	DCI	0.531 ± 0.075	0.178 ± 0.032 **	0.238 ± 0.040	0.471 ± 0.163 ^#^
	MI	17.248 ± 1.382	18.102 ± 0.653	18.864 ± 1.003	18.435 ± 1.125
	DCI/MI (%)	3.083 ± 0.333	0.974 ± 0.160 **	1.282 ± 0.234	2.599 ± 0.911 ^#^
Bone marrow	DCI	0.052 ± 0.009	0.054 ± 0.010	0.098 ± 0.009	0.167 ± 0.026 ^###^
	MI	4.290 ± 0.319	5.432 ± 0.459	4.916 ± 0.417	4.498 ± 0.441
	DCI/MI (%)	1.142 ± 0.214	0.987 ± 0.153	2.095 ± 0.267 ^#^	4.100 ± 0.898 ^###^

^$^ Serum inositol content expressed as μg/mL serum, other expressed as μg/mg protein. The Sham group mice were treated with the vehicle (Sham). The mice with diabetic osteoporosis were treated with the vehicle (DO), 50 mg pinitol/kg Bw/day (DO + LP), and 100 mg pinitol/kg Bw/day (DO + HP) for 5 weeks, respectively. Values were expressed as mean ± SEM, *n* = 7. * *p* < 0.05, ** *p* < 0.01 vs. Sham; ^#^
*p* < 0.05, ^###^
*p* < 0.001 vs. DO.

## Data Availability

Not applicable.

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
