# Peer review of "D-Pinitol Improved Glucose Metabolism and Inhibited Bone Loss in Mice with Diabetic Osteoporosis"

_molecules, 2023, doi:10.3390/molecules28093870_

Round 1
Reviewer 1 Report
This work revealed that pinitol improved glucose metabolism and inhibited bone loss in DO mice via elevating DCI levels in tissues. The result is interesting. The points for improvement are as follows:
1. The discussion was not thorough enough, for example, to discuss possible pharmacological mechanisms.
2. The format of the references is not uniform.
Minor editing of English language required.
Author Response
Dear reviewer,
Many thanks for your valuable comments and suggestions.
- The discussion was not thorough enough, for example, to discuss possible pharmacological mechanisms.
As the reviewer pointed out, we did not give enough discussion about the mechanism of D-pinitol. Thus, we added more information regarding the possible pharmacological mechanism of pinitol (Line 275-277, 294-306 in the revised MS). As a matter of fact, the protective mechanism of D-pinitol and DCI on either diabetes or osteoporosis have been reported separately, and both pinitol and DCI manifested a high consistency when they were exerting their anti-diabetic or anti-osteoporosis effect (Line 294-304 in revised MS). Therefore, one of the major purposes of this study is to prove the conversion of pinitol to DCI in vivo, which is supposed to be crucial for pinitol in diabetic osteoporosis treatment.
- The format of the references is not uniform.
We have checked the format of the references again and corrected them.
Thank you again for your comments and suggestion; we are willing for any further discussion.
Best regards,
Xinxin Liu,
Department of marine bioscience,
Tokyo University of Marine Science and Technology,
4-5-7 Konan, Minato City, Tokyo,
Japan
Email address: d221020@edu.kaiyodai.ac.jp
Reviewer 2 Report
The authors of this study investigated the potential therapeutic effects of D-pinitol on fasting glucose levels and bone loss in diabetic osteoporosis mice. The study found that D-pinitol treatment improved glucose metabolism, as evidenced by a decrease in blood glucose levels and an increase in insulin sensitivity. Additionally, D-pinitol treatment prevented bone loss in diabetic osteoporosis mice, as indicated by decreased serum bone biomarkers, and improved dry femur weight and bone mineral content. These findings suggest that D-pinitol has the potential as a therapeutic agent for diabetic osteoporosis. The authors provided solid evidence to support their conclusion. Regarding some minor comments:
1. Although osteoporosis is more common in postmenopausal women, men with diabetes can also develop osteoporosis. Are there any animal models for studying male diabetic osteoporosis?
2. In Figure 2B, since the food intake of DO+LP and DO+HP groups did not differ from the DO group, how did pinitol decrease blood glucose levels?
3. The study did not specifically measure bone strength using mechanical testing methods. Did the authors observe whether there is any difference in bone strength among the groups?
Author Response
Dear reviewer,
Many thanks for your valuable comments and suggestions. Following are our answers,
- Although osteoporosis is more common in postmenopausal women, men with diabetes can also develop osteoporosis. Are there any animal models for studying male diabetic osteoporosis?
The rapid decline of estrogen levels in postmenopausal women is one of the major factors in the pathogenesis of diabetic osteoporosis. Therefore, we performed an ovariectomy to accelerate the establishment of the diabetic osteoporosis model. However, the long-term of diabetes itself induces bone loss as well, which can be observed in either gender. Besides, differing to females, secondary osteoporosis (ex. Diabetic osteoporosis) is the major cause in males [1]. Therefore, to study male diabetic osteoporosis, it is advisable to employ certain male diabetic mice with long-term diabetes conditions (ex. male KK-Ay mice), which has been applied to research [2].
- In Figure 2B, since the food intake of DO+LP and DO+HP groups did not differ from the DO group, how did pinitol decrease blood glucose levels?
In this study, the measurement of blood glucose levels was performed after 16 h of fasting every time (Fig 3A). Therefore, in our opinion, food intake would not influence the fasting blood glucose levels. Generally, pinitol was supposed to regulate glucose metabolism in two possible ways. (i) Improve insulin resistance. (ii) Improve pancreatic function. Given that the DO mice model we established has an impaired pancreatic function but no insulin resistance, we prefer to hold the view that pinitol suppressed DO mice via an improvement of pancreatic function (Fig 3C, Line 275-277 in revised MS).
- The study did not specifically measure bone strength using mechanical testing methods. Did the authors observe whether there is any difference in bone strength among the groups?
Exactly, as the reviewer pointed out, mechanical testing would give more comprehensive information about bone strength and microstructure. Unfortunately, however, because of the lack of specific equipment in the lab, it was difficult for us to perform these measurements. But, we supposed that the indicators we measured, including dry bone weight and cancellous bone rate, could reflect the changes in bone strength from the side. Since the inner cancellous bone plays an important role in maintaining bone strength, an inhibition of cancellous bone loss by a pinitol treatment was speculated to be related to the improvement of bone strength.
Thank you again for your comments and suggestions; we are willing for any further discussion.
[1] Issa, Claire, Mira S. Zantout, and Sami T. Azar. "Osteoporosis in men with diabetes mellitus." Journal of Osteoporosis 2011 (2011).
[2] Takahashi, Asako, et al. "Erucic acid-rich yellow mustard oil improves insulin resistance in KK-Ay mice." Molecules 26.3 (2021): 546.
Best regards,
Xinxin Liu,
Department of marine bioscience,
Tokyo University of Marine Science and Technology,
4-5-7 Konan, Minato City, Tokyo,
Japan
Email address: d221020@edu.kaiyodai.ac.jp